# How Sanctuary Chimpanzees (*Pan troglodytes*) Use Space after Being Introduced to a Large Outdoor Habitat

**DOI:** 10.3390/ani13060961

**Published:** 2023-03-07

**Authors:** Amy Fultz, Akie Yanagi, Sarah Breaux, Leilani Beaupre, Nick Naitove

**Affiliations:** 1Chimp Haven, 13600 Chimpanzee Place, Keithville, LA 71047, USA; 2Office of Academic Affairs, Niagara County Community College, 3111 Saunders Settlement Rd, Sanborn, NY 14132, USA; 3Department of Veterinary Resources, University of Louisiana at Lafayette-New Iberia Research Center, P.O. Box 13610, New Iberia, LA 70562, USA; 4Independent Researcher, Tumwater, WA 98512, USA; 5WildThink, P.O. Box 1422, Wheat Ridge, CO 80033, USA

**Keywords:** space use, indoor/outdoor enclosures, species-typical behavior, welfare

## Abstract

**Simple Summary:**

Chimpanzees live in large groups in large territories in the wild, although this is less common in captivity. At Chimp Haven, the world’s largest chimpanzee sanctuary, the chimpanzees are integrated into larger groups and provided access to large, forested areas that encourage species-typical behaviors. In this study, we observed 18 chimpanzees as they were integrated into a large group and introduced to new environments over a 7-month time frame. Changes in the locations where the chimpanzees spent their time and conspecifics whom they spent their time with were recorded during both daytime and nighttime observations. We examined the changes in the chimpanzees’ spatial use of indoor and outdoor areas as well as arboreal and terrestrial locations. Overall, the chimpanzees’ use of space and proximity to others changed over time.

**Abstract:**

Wild chimpanzees live in large, mixed-sex groups that display a fission–fusion social organization. To provide a social environment more like that of wild chimpanzees, Chimp Haven integrated smaller groups of 3–4 individuals into one large group of 18 individuals. This large group was introduced to a 20,234.28 m^2^ forested habitat and associated indoor areas. This space was designed to allow the individual chimpanzees to choose their proximity to social companions and provide the group with a more natural environment in which they could express more species-typical behavior. The study took place over a 7-month period that began two weeks prior to the first groups being integrated and ended 4 months after the chimpanzees were released into the habitat. We collected data on the chimpanzees’ arboreal, terrestrial, indoor, and outdoor spatial use. The chimpanzees’ proximity to their nearest neighbor was also recorded, noting whether they were touching or within arm’s reach. Data were collected during daytime and nighttime hours and were utilized to make management decisions about potential group formations. We examined the data using generalized linear mixed models (GLMMs) with individual subjects as a random factor and months as a repeated measure for location and proximity results. There were significant differences in the use of arboreal and outside space over the 7-month time frame, with females more likely to use the arboreal space than males. The chimpanzees were more likely to utilize the habitat over time and increased their proximity to group mates. The results of this study indicate that the chimpanzees responded positively to living in large groups in a spacious naturalistic environment.

## 1. Introduction

In the wild, chimpanzees live in large flexible groups of 20 to 100 or more individuals that occupy territories typically ranging from 5 km^2^ to 27 km^2^, although some territories may even extend to 300 km^2^ [1,2,3,4,5]. Within these large territories and the fission–fusion social organization of their groups, individuals climb, travel, rest, make nests, utilize tools, and forage for food. They also interact with each other through mating, cooperative hunting, foraging, grooming, and patrolling the boundaries of these territories. Wild chimpanzees use both arboreal and terrestrial spaces within their habitats. In the wild, chimpanzees of both sexes also typically spend their time in proximity to familiar group mates [6,7]; however, they may distance themselves from unfamiliar chimpanzees or even seek out and attack strangers [8,9,10]. The environment of wild chimpanzees is different from typical North American climates in that they live in climates near the equator where the temperatures remain warmer and relatively steady throughout the year [11,12], whereas North American climates experience seasonal fluctuations where temperatures range from cold in the winter months to hot and humid in the summer months. 

In captive environments, most United States facilities that house chimpanzees are required by the Animal Welfare Act to adhere to the Guide for the Care and Use of Laboratory Animals [13], which specifies the minimum amount of space required for chimpanzees. Recommendations for adult chimpanzees living in pairs or groups is a minimum floor space area greater than or equal to 2.32 m^2^ per chimpanzee. There are no recommendations for outdoor space. More recently, informed by Else [14], the United States National Institutes of Health recommended that chimpanzees should have 23 m^2^ per individual; however, Else also suggests that more research is necessary to determine a more definitive answer to the question of what optimal space is for captive chimpanzees.

Many captive facilities exceed these requirements, especially in regard to expanded and more complex outdoor space, which may provide a more species-typical environment [15,16]. In captive settings, we may want to further magnify available space to provide additional opportunities for species-typical social groupings and behavior. An important component to increasing space and complexity is understanding the behavioral changes of chimpanzees in the expanded environments and how this may contribute to future decisions concerning the captive care of chimpanzees.

Increased access to larger, more complex environments has been associated with positive behavioral changes. These changes include increases in species-typical behavior such as climbing and locomotive behaviors [17,18], prosocial behavior [15,19], increased traveling time [16,17], and expanded vegetation use [18]. Species-typical activities such as nest building, and ant fishing are also evident in more complex spaces [20,21,22]. After moving a group of chimpanzees from a small indoor environment to a larger indoor/outdoor environment, one group of researchers found an increase in locomotion and climbing, but a decrease in feeding behaviors [16]. Ross et al. [23] found a reduction in abnormal and attention-seeking behaviors when zoo chimpanzees moved from a non-naturalistic indoor enclosure to a more naturalistic indoor/outdoor enclosure. In another study, when zoo-housed chimpanzees had access to their outdoor enclosures, their daily travel distance increased by 92% [17].

In contrast to generally positive behavioral changes resulting from expansions of space, more complex behavioral changes may occur when space is reduced. For instance, male and female chimpanzees may employ different behavioral strategies for long-term and short-term reductions in available space that lead to decreased aggression in both conditions. Notably, in one study, males increased affiliative behavior in both conditions, while females increased their affiliative behavior only during the long-term condition [24]. Increased aggression and self-directed behaviors have also been associated with moving to smaller, more confined spaces [19]. Despite some negative behaviors associated with decreased space, chimpanzees do not always utilize all available space when given the option. Ross et al. [15] found that chimpanzees housed in a naturalistic indoor/outdoor enclosure spent less than half of their time (33.2%) outdoors when given the choice over a 4-year period. They also noted that the portion of the enclosure used inside did not change significantly when the chimpanzees had access to the outdoor portion of their enclosure; they spent half their time in only 4.2% of the available space, including vertical space. The authors of the study acknowledge that the use of the space does not necessarily explain the importance of allowing the chimpanzees’ access to larger, more complex enclosures that would provide choices in behavioral responses such as avoidance and escape.

Many factors influence patterns of space utilization in captive primates including the quantity of the space involved, the complexity of the enclosures, and the functionality of the available space [15,25,26,27,28,29]. The presence of environmental features, such as mesh and solid concrete walls, corners, climbing structures, and substrates, may also affect space use. Characteristics of these features are often important. One study suggested that proximity to keeper areas might have increased the time the chimpanzees spent in the area [27]. Other characteristics, such as whether the substrate consists of grass, straw, woodchip bedding, or concrete floors [23], the nature and layout of enclosure vegetation [18], and the height and material (natural or artificial) of climbing structures [15], may also affect space use.

For captive chimpanzees, social pressures, age, sex, rearing history, and group composition may be additional factors that influence the use of space [30,31]. As with chimpanzees in the wild, dominant females tend to monopolize preferred sites [31] and younger chimpanzees avoid spaces occupied by more dominant or older chimpanzees [29]. Captive chimpanzees have their social companions chosen for them, which may lead to additional social pressures. Several studies have also suggested that captive chimpanzees prefer elevated areas [3,15,16,23,27]. Ross and Lukas [27] discovered that the highest rates of enclosure use were at elevations near the ceiling, even with available access to many other levels of elevated surfaces. Another group of captive chimpanzees spent almost three-fourths of their time (73.4%) above ground level, utilizing elevated structures most frequently [16]. This pattern of space use closely mimics that of wild chimpanzees, which spend from 33 to 80% of their time in arboreal locations depending upon the location and forest structure [3,32]. 

While elevation is important to a chimpanzee’s use of space, social pressures may also influence location. The fission–fusion group movements observed in wild chimpanzees would suggest that captive chimpanzees would also disperse throughout their enclosures. However, at the Edinburgh Zoo, when provided with a large, complex enclosure (2332 m^2^ indoor/outdoor exhibit comprising multiple interconnected areas), chimpanzees chose to maintain close proximity to members of their subgroup rather than disperse individually [33]. Thus, when given access to larger and more naturalistic environments, captive chimpanzees may not maximize the use of available space, choosing instead to cluster in specific areas to maintain proximity to group members or to utilize preferred spaces such as elevated structures [34]. Ross et al. [15] suggest that having the choice to use the space alone can be a positive enrichment factor for primates, even when they do not use it.

Although there are many studies related to the use of space by captive chimpanzees in zoos and research facilities, it is still unclear how sanctuary chimpanzees utilize their space. The space available to chimpanzees in sanctuaries may differ from that provided in either of the other two settings. For example, although it has become more common for zoos to provide their chimpanzees with larger, more naturalistic areas that include some trees, we are unaware of any multi-acre outdoor-forested habitats in research facilities or zoos. Another difference in the zoo setting would be the chimpanzees’ exposure to the public, including public viewing areas that may or may not affect their use of space [35]. Similarly, all captive chimpanzees may associate certain areas with sedations or other procedures, which could affect their use of space.

To promote a full range of natural behaviors in the resident sanctuary chimpanzees, spacious and enriched spaces are included as part of the overall behavioral management plan at Chimp Haven, the world’s largest chimpanzee sanctuary, located in Keithville, LA, USA. Chimp Haven forms large social groups to provide the chimpanzees with opportunities to engage in species-typical social behaviors within the enriched spaces available to them. The current study was conducted at Chimp Haven to evaluate the chimpanzees’ behavior before and after they were given access to a new large, forested habitat.

We hypothesized that the chimpanzees would show changes in their behavior after being introduced to the habitats. Specifically, we predicted that (1) the amount of time the chimpanzees spent outdoors in the habitat would increase over time as they became more comfortable with their new surroundings, and (2) that they would spend more time in arboreal locations due to the increase in available space and additional opportunities for climbing provided by the forest. The habitats at Chimp Haven are mostly loblolly pine trees (*Pinus taeda*), small conifers with brittle lower limbs, which often break and cannot bear much weight. Therefore, we predicted (3) that the females would use the arboreal space more than males due to their smaller weights [36,37,38].

Additionally, we investigated the relationship between an individual subject’s location and proximity to conspecifics to monitor the group’s social structure development over time via the assessment of changing relationships within it as the space available to them increased and became more complex [39,40,41]. We predicted that (4) the chimpanzees would spend more time in proximity to other group members over time, (5) particularly unfamiliar group members, as the group stabilized. We also predicted that (6) our oldest chimpanzees, whom we suspect or know were born in the wild, would utilize the habitat to a greater degree. Specifically, (7) we expected the wild-born individuals to make greater use of the arboreal space in particular. We anticipated more arboreal activity by wild-born chimpanzees because they would have had access to large trees in the central African landscape for climbing and nesting [3,42], as well as prior work conducted at Chimp Haven and other facilities [21,22,43]. 

## 2. Materials and Methods

### 2.1. Facilities

The chimpanzees had access to six indoor bedrooms (each 192 f^2^/17.8 m^2^) throughout the study, which were always ventilated and heated during colder weather. Each bedroom contained toys, hammocks, perches, and a skylight, as well as materials for nesting, including hay, pine straw, blankets, and wood chips. The bedrooms connect to each other and to various outdoor enclosures through a series of doors and chutes.

At the onset of the study, in May 2005 (Month 1), the chimpanzees had access to these bedrooms and a circular outdoor open-topped corral. The corral was a 5776 f^2^ (536.6 m^2^) enclosure contained by 17 f (5.2 m) high poured concrete walls with mesh windows at eye level. The corral had natural grass ground covering and wooden climbing structures.

In July 2005 (Month 3), after additional introductions of new individuals to the existing group, the chimpanzees were given access to an extensive forested outdoor habitat in addition to the round corral and bedrooms. The habitat space was novel to all the chimpanzees in the group. The habitat measured 217,800 f^2^ (20,234.28 m^2^) and was a natural temperate mixed broadleaf and conifer secondary forest with natural ground covering. The area was contained on two sides by 17–20 f (5.2–6.1 m) high poured concrete walls with mesh windows at eye level (5 ft (1.52 m). All trees were cleared within 20 f (6.1 m) of the walls and small trees [less than 5 in (12.7 cm) in diameter] were cleared within 40 f (12.2 m) of the walls to prevent egress. The third side of the habitat was bound by a water moat, which measured approximately 60 f (18.3 m) across (Figure 1). The moat’s construction included a gradual slope to 8 f (2.4 m) deep in the center, a clay liner, and a dedicated well used to pump water to the moat if needed. Safety precautions included a 10 f (3 m) wide shallow water area of approximately 1 f (0.3 m) depth of water adjacent to the shore of the habitat. This area was contained with a concrete weir wall at the water level. The weir wall had a cable along the top and chain link fencing placed under the water to assist a chimpanzee in getting out of the water should they fall in.

The chimpanzees had access to the corral, indoor bedrooms, and the habitat at all times, except during sanitation of the enclosures. Chimpanzees were not permitted outside access to the habitat during the occasional instances of severe inclement weather. Indoor bedrooms were cleaned daily, and cleaning of outdoor enclosures occurred every two weeks. During cleaning or inclement weather, chimpanzees were housed in their indoor bedrooms. The chimpanzees at Chimp Haven have the option of sleeping outdoors in the habitat with access to natural vegetation, or indoors in a heated and ventilated area with hammocks and other nesting materials such as blankets and hay.

The chimpanzees were fed a commercially available primate diet and fruits and vegetables twice per day. A forage mix (including various seeds, nuts, and cereals) was scattered in the animal areas three times each week. Enrichment, which included both food and non-food items, was provided daily, seven days per week. In the habitats, in addition to the regular diet, natural vegetation was also available for consumption by the chimpanzees.

### 2.2. Subjects

The subjects were 18 adult chimpanzees (n = 11 females, 7 males) aged 16 to 46 years (mean = 30.94) retired from biomedical research between April and May 2005. There were 7 wild-born chimpanzees in the group (n = 6 females, 1 male) who were all raised by wild-born mothers for unknown periods of time (Table 1).

### 2.3. Data Collection

This study was conducted from May 2005 (Month 1) through December 2005 (Month 7); however, the data collection was not continuous; no data were collected in the month of September due to staffing constraints. More specifically, data collection began two weeks prior to the formation of the first sub-groups and was conducted by nine trained observers. 

We did not measure inter-observer reliability (IOR) for this study but focused on using staff trained at the facility who had undergone specific training and passed required tests to be able to take data reliably. These requirements included positive identification of individual chimpanzees, passing a test on the ethogram used for this study, and general husbandry and behavior information on chimpanzees. The first author trained all other observers on how to record data for this study. As resources limited our ability to focus on IORs, we instead focused on the practical applications of recording these data.

We conducted this study in two phases over a total of seven months in 2005: from May through August (Months 1–4) and from October through December (Months 5–7). Both daytime (8:30 a.m.–4:30 p.m.) and nighttime (6:00 p.m.–6:00 a.m.) observations were conducted using instantaneous or scan sampling [44] at each hour between 6:00 p.m. and 12:00 a.m., at 5:00 a.m. and 6:00 a.m., and at each hour between 8:30 a.m. and 4:30 p.m. Chimp Haven employs nighttime caregivers, so the nighttime data were collected directly and not via the use of cameras or other technology. 

Due to the concerns of the animal care staff that the chimpanzees’ sleep was being disrupted, we eliminated one data-point, the 11 p.m. check, after the first week. This was performed only after reviewing the previously collected data and determining that the chimpanzees were typically in the same place during most nighttime checks. We removed all 11 p.m. data-points from the first week’s data for subsequent analysis.

For each observation, we noted all subject-locations indoors or outdoors including elevation (arboreal or terrestrial). The outdoor condition included both the outdoor corrals and the habitat. We defined terrestrial as a ground-level location for the observed chimpanzee and included indoor and outdoor substrates. We defined arboreal as any above-ground-level location, which included perches, shelves, fire hose hammocks, trees, and permanent wooden structures. Potentially ambiguous body-posture or positioning (e.g., where only one hand or foot was elevated) was resolved by further specifying location as the position of three-quarters of the observed chimpanzee’s body. We also recorded proximity, which was defined as being within arm’s reach of another chimpanzee, as well as contact which was defined as touching another chimpanzee. Proximity included both when the chimpanzees were engaged in active social interactions and when they were passively within arm’s reach or touching another group member. Proximity and contact were combined for data analysis. 

We defined ‘unfamiliar’ group members as those who had spent time together as group mates for less than or equal to 6 weeks. ‘Familiar’ members were those who had been group mates for more than 6 weeks. These definitions were based on staff input and observations regarding when the groups appeared stable, with only superficial wounding, affiliative interactions, and reduced aggression, as well as the logistics of the introductions which required us to conduct the integrations over time. 

### 2.4. Data Analysis

For all analyses, we first calculated the percentage of scans in which each subject was in a particular condition; indoors, outdoors, arboreal, and terrestrial. We then determined proximity to familiar and unfamiliar group members over time. To determine the use of outdoor space as a proportional percentage, we divided the number of the relevant scans by a total number of monthly scans per subject (and multiplied the number by 100) because the chimpanzees could be outdoors in the corral as well as the habitat.

To calculate the use of the habitat space, we counted all scans in which the subject was in the habitat and divided the number of scans by the number of monthly scans in which they were in any outdoor locations. For indoor arboreal locations as well as terrestrial locations in the habitat, percentages were calculated by counting the number of scans in which the subject was located arboreally in indoor locations (or terrestrially in the habitat) and dividing that number by a total number of scans in which the subject was found indoors (or in the habitat). 

For proximity calculations, we calculated percentages by counting the number of scans in which the subject was within arm’s reach of at least one group member and divided the number by a total number of monthly scans involving those of all possible neighbors (i.e., the number of monthly scans x 17 possible partners). This calculation method, as opposed to a proper dyad calculation, was employed because of our lack of directionality data; therefore, we acknowledge that there is a potential risk of duplication in our proximity data. The percentages for the analysis of proximity to unfamiliar conspecifics were calculated by counting the number of scans involving ‘unfamiliar’ neighbors and dividing the number by the total number of scans in which a subject was found with at least one other group member.

#### Statistical Analysis

For all analyses, we conducted generalized linear mixed models (GLMMs) with the beta distribution to account for the proportional (percentage) response variable, using the glmmADMB package in R 3.6.3 created by R Core Team, Vienna, Austria (2020). For all predictions, we conducted generalized linear mixed models (GLMMs) with the beta distribution due to the proportional (percentage) response variable. When the response variable in a dataset included exact zero values, we universally added the value of 0.01 to the entire dataset to fit the data for a beta-distributed GLMM, which does not accept the values of exact zero or one. The datasets for habitat, all arboreal space, indoor arboreal space, and habitat terrestrial space were included. Each GLMM included subject as a random factor, and sex, age (continuous), rearing history, and time (repeated measure) as fixed factors. For the variable ‘time,’ we used the month of May as a reference point for the predictions regarding outdoor and arboreal locations. July was our reference point for other predictions, since this was when the group had access to the forested habitat for the first time. All models were initially examined with potential four-way and three-way interactions; however, the dataset of 18 subjects could not yield any results with these complex interactions. The variable time was kept as a single factor in all models due to our main hypothesis to identify the effect of time in space use, and the continuous variable age was also kept as a single factor. Consequently, we ran all potential, feasible models with the interaction term of sex*rearing history, and then without them to select the best model based on the smaller value of Akaike Corrected Information Criterion (AIC) and the zero value of Delta AIC (Supplement 1). The best fit model with the Delta AIC value of zero is presented for each prediction in the Section 3 below. 

We assessed the significance of each overall model against the null model based on likelihood ratio tests with the ANOVA function and chi-square distribution. Because we analyzed the same set of data for seven predictions, we applied the Bonferroni correction to set critical levels of statistical significance for the overall models for each prediction; the resulting critical value was *p* ≤ 0.05/7 = 0.0071. A probability of 0.05 < *p* ≤ 0.1 was considered a non-significant tendency.

## 3. Results

A total of 35,756 (SD ± 6.6) scans were collected during the entire study (May: n = 3123; June: n = 7542; July: n = 6425; August: n = 5328; October: n = 4842; November: n = 4500; December: n = 3942). We collected a total of 1988 scans for 17 subjects, and 1960 for one subject. Our analyses resulted in significant results in multiple categories. First, we discuss the main results in each category and then we discuss the complete results in more detail (Table 2). 

### 3.1. Use of All Outdoor Space

The best fit model included only the variable time (χ^2^(6) = 115.45, *p* ≤ 0.0005). Time was significantly associated with the subjects’ use of outdoor locations (Table 2). Compared to May, subjects were more likely to be outdoors in the months of June, July, August, October, November, and December. Compared to July, the month the chimpanzees accessed the habitat for the first time, subjects were significantly more likely to use the habitat in the months of October, November, and December (Figure 2).

### 3.2. Habitat Use

The best fit model for the use of the habitat included the variables time, age, and rearing history (χ^2^(7) = 104.48, *p* ≤ 0.0005). Each of these variables was significantly associated with the chimpanzees’ use of the habitat (Table 2). Subjects that were wild-born and reared by a wild mother were more likely to use the habitat than were captive-born or nursery-reared subjects (Figure 3). Age showed the weakest significance in the model (Table 2). Nevertheless, the percentage of the scans showing the subject’s usage of the habitat was higher among older (geriatric) subjects compared to their younger (non-geriatric) counterparts, in all the months examined except for November (Figure 4). In Figure 4, the age variable is divided into two categories, non-geriatric (younger than 35 years old) vs. geriatric (35 years old or older) to provide a context. These categories were chosen based on multiple prior studies related to chimpanzee longevity and mortality [45,46,47,48]. 

### 3.3. Use of Arboreal Space

The best fit model for the use of arboreal space included time and sex (χ^2^(7) = 63.995, *p* ≤ 0.0005). Subjects’ sex was significantly associated with their overall usage of arboreal locations with females more likely to be arboreal than males (Table 2), and subjects’ overall use of arboreal locations increased in every month after May (Figure 2 and Figure 5).

### 3.4. Use of Indoor Arboreal Space

The best fit model for the use of indoor arboreal space included the variables time and sex (χ^2^(7) = 40.43, *p* ≤ 0.0005). Time was associated with the subjects’ usage of indoor arboreal locations; compared to May, subjects were more likely to use indoor arboreal locations in July through December (Table 2, Figure 6). Males were less likely to be arboreal in indoor spaces compared to females (Table 2). 

### 3.5. Use of Terrestrial Space (Habitat)

The best fit model only included the variable time (χ^2^(4) = 27.3, *p* ≤ 0.0005). The subjects were more likely to be terrestrial in the habitat during the later months of October through December, compared to July (Table 2, Figure 7). 

### 3.6. Proximity to All Group Members

The best fit model for the subject’s proximity to at least one group member included variables time and sex (χ^2^(5) = 58.76, *p* ≤ 0.0005). Compared to July, subjects were more likely to be within arm’s reach of at least one group member in the months of November and December, while October showed a nonsignificant tendency (Table 2, Figure 8). Males were less likely to be in proximity to another group member compared to females (Table 2). 

### 3.7. Proximity to Unfamiliar Group Members

The best fit model included variables time and sex (χ^2^(5) = 15.16, *p* = 0.0097). This overall model was a nonsignificant tendency after applying the Bonferroni correction (i.e., *p* ≤ 0.0071). Nevertheless, subjects were more likely to be within arm’s reach of an unfamiliar group member in December compared to July (Table 2). Males were more likely to be within arm’s reach of an unfamiliar group member than females (Table 2, Figure 9).

## 4. Discussion

### 4.1. Use of All Outdoor Space

In this study, sanctuary chimpanzees regularly used their outdoor areas in specific ways, particularly when temperatures were milder. Summers in Northwest Louisiana, where Chimp Haven is located, can reach temperatures in the upper 90s (F) (32.2 °C) with high humidity. Chimp Haven provides fans, air vents, and covered areas for the chimpanzees, providing shade and potentially cooler inside locations. In comparison, low temperatures may be in the 30s (F) (−1.11C) between the months of October and December (https://www.wunderground.com/weather/us/la/shreveport; accessed 12 February 2023). Specifically, in 2005, the year this study was conducted, the average monthly temperatures were: May: 72.7F (22.6C); June: 82.03F (27.8C); July: 82.69F (28.15C); August: 84.43F (29.13C); September, when no data were collected: 82.24F (27.94C); October: 66.87F (19.37C); November: 60.05F (15.58C); and December: 48.76F (9.32C). 

This differs from some previous studies of zoo-living apes who were reluctant to utilize new outdoor enclosure space [15,49]. However, Ross [15] determined that chimpanzees and gorillas both used their outdoor areas, with chimpanzees using the area for about a third of the observation time, while Bloomsmith [50] found that research facility chimpanzees spent 45% of their time outside. 

Over time, the chimpanzees in this study spent more time outside, particularly in the habitat, and less time inside, increasing their exploration of the forested habitat, especially during the cooler months of October to December. Both wild and captive chimpanzees are adaptable and live in many types of environments. In the wild, chimpanzees in different communities have developed different strategies for hunting, grooming, and ant and termite fishing, as well as different group compositions in order to adapt to different environments [51,52]. In captivity, chimpanzees live and breed in various areas around the world where outdoor ambient temperatures and enclosure sizes vary dramatically [53]. Given this adaptability, we would expect to see a difference in the use of areas based on external environmental factors such as temperature and weather. In this study, we found that the chimpanzees used outdoor space less when temperatures were highest. We suspect this may be due to seeking cooler temperatures and covered and shaded areas in their indoor enclosures. Indoor bedroom areas also have treated concrete floors and mesh perches which may provide cooler resting spaces than outdoor areas warmed by the sun. However, multiple variables related to weather, such as precipitation, wind speed, humidity, and cloud cover, may also play a role in the chimpanzees’ use of space and would be interesting factors to include in further studies. Although the weather may have played a role in the use of the outdoor space, the authors also believe that the increased use of the area over time may have been related to the novelty of the space wearing off as the chimpanzees became more familiar with their new area, its terrain, and features. The chimpanzees had just arrived at Chimp Haven in the spring of 2005, and, therefore, all the areas were new to them. Familiarity and feeling more at ease with their new groupmates may also have contributed to their use of outdoor space. 

Understanding how sanctuary chimpanzees use their space has important implications as new enclosures are built at various sanctuaries and zoos, with additional chimpanzees being moved to these facilities. Housing and ongoing care for large populations of chimpanzees costs millions of dollars each year [54]; therefore, it is important to determine types of enclosures that are utilized the most by the chimpanzees. It is also important to define what types of space, both in terms of size and complexity, should be considered optimal to maximize chimpanzee welfare.

### 4.2. Habitat Use

The chimpanzees in this group immediately interacted with their spacious and complex forested habitat and increased their use of the area over time. Certainly, in this study, there were individual chimpanzees who used the forested space more than others did. For example, chimpanzees born and reared in the wild were more likely to use the habitat than captive-born, nursery-reared chimpanzees. At Chimp Haven, we have witnessed captive-born chimpanzees learning to climb trees by observing older, wild-born chimpanzees [21]. Older chimpanzees were also more likely to use the habitat than their younger counterparts; however, it should be noted that the oldest chimpanzees were also wild-born. We suggest that future research specifically on captive-born chimpanzees of different age groups might assist in elucidating this result [55]. In another study, young chimpanzees used enriched outdoor enclosures more frequently than their older counterparts, 72% of the time [29], which is the opposite of our findings. Therefore, given that the wild-born population of chimpanzees in captivity is declining and the majority of captive chimpanzees are adults, additional studies may assist us in determining what is optimal for the current captive population.

### 4.3. Use of Arboreal Space

The chimpanzees’ use of space was also altered depending upon which areas the chimpanzees had access to. When given increased space and additional opportunities to climb in the habitat, the chimpanzees spent more time in arboreal locations, suggesting more broad use of the space available to them. In the wild, chimpanzees can spend as little as 3% of their time off the ground but are more likely to spend 50–80% of their time arboreally depending on the population of chimpanzees and the composition of the local forest [3]. Differing climates, tree availability, human influence, and predators may also influence arboreality [42,56]. Because of this, we might expect chimpanzees to spend much of their time in a forested habitat at an arboreal elevation, and the current results suggest that females did spend more time arboreally while in the habitat. Chimpanzees retiring to a sanctuary may not have had the opportunity to climb native trees before, and, therefore, may be reluctant to climb what may appear to them to be an unstable and risky structure. The weights of the chimpanzees may also have contributed to their reluctance to climb the trees, particularly for the larger males. In the wild, at Mahale Mountains National Park, males averaged 42.0 kg and females averaged 35.2 kg [38]. In captivity, these differences may be even more extreme; in a recent publication, adult male and female chimpanzees in various settings ranged from 43 kg up to 72 kg [36]. In any setting, adult males typically weigh more than the adult females. This may make males more hesitant to use arboreal spaces that do not offer a sturdy support system, such as the small loblolly pine trees that are the predominant species in the habitat at Chimp Haven. Loblolly pine trees have also naturally evolved to drop their lower limbs as they age, leading to limbs that cannot support the weight of a chimpanzee closest to the ground. In addition, all loblolly pines have a specific gravity of 0.51, meaning that the trees are neither durable nor strong (https://edis.ifas.ufl.edu/publication/ST478; accessed 21 January 2023). Indeed, there are multiple cases of wild chimpanzees falling to their deaths from the trees [57].

### 4.4. Use of Terrestrial Space (Habitat)

The chimpanzees in this study were more likely to be found in terrestrial locations during the months of October to December when the trees are losing their leaves and there are cooler temperatures. The authors have witnessed the chimpanzees climbing trees and foraging for edible forest fruits and leaves in the spring and summer months but this has rarely been recorded during the fall and winter months at Chimp Haven [21]. The forest floor does provide thick vegetation, visual barriers, and multiple options for tool use and foraging. These factors may have influenced the group’s increased use of terrestrial locations when in the habitat. Captive chimpanzees are known to spend increased amounts of time foraging and exploring when substrate or bedding is provided. Many facilities in the United States include this as part of their environmental enhancement and behavioral management plans [43,58]. The forest floor and vegetation of the habitat provides a nearly endless substrate for exploration and foraging.

### 4.5. Proximity to Group Members 

In this study, the chimpanzees were significantly more likely to be within arm’s reach of a group member over time—this did not vary with the familiarity of the group mate. In other words, the chimpanzees tended to be in proximity to one another more often in October and were significantly more likely to be in proximity in November and December. In another study on this same population, both male and female chimpanzees increased their proximity to group mates after being released into the habitat [59]. Proximity to group mates may change due to available space, housing conditions, or familiarity with group members. We could not determine whether or not this was due to the new group becoming more stable over time or the increase in space as both occurred concurrently. 

During the course of our observations, proximity as a proxy of familiarity between chimpanzees was not definitive but did show a tendency towards increased association. Given the fact that available space was smaller at the beginning of the study, and we might expect the chimpanzees to have been more likely to be in proximity due to the reduced space, we do see the increase in proximity over time, with the additional access to the large habitat, to be a sign of improved relationships or associations. According to one study, having the choice of where to be or whom to be with, particularly when multiple options are provided, is more important than the amount of space available or total density [60].

### 4.6. Future Considerations 

In the future, it would be interesting to see if a more stable large group would continue to use the space in the same way or if the use of the space and proximity to group members would change over time. Recording behavioral information on species-typical activities in the habitat may also be important to our characterization of the use of space. In another study on space use, behavioral diversity increased with changes in space [58]. This study did not record such behaviorally diverse activities such as ant fishing, tool caching, hunting, consorts, or patrolling of the habitat, which are all species-typical behaviors that we have observed in the Chimp Haven habitat (first author, personal observations). Additionally, other behaviors such as grooming, nest building, and tool use may also be indicators of positive welfare and flexible adaptation on the part of the chimpanzees [43]. Recording the presence or absence of abnormal or atypical behaviors while the chimpanzees are in the habitat or have access to it, as well as examining wounding levels, and markers of physical and psychological well-being such as hair loss and body mass, may also contribute to our understanding of the impact of the use of this space on chimpanzee welfare [43,58].

This study’s initial characterization of chimpanzee space use at a sanctuary within a naturalistic forested habitat illustrates how larger, more natural enclosures may affect captive chimpanzee space use over time. This study, therefore, is likely to have implications for the design of future chimpanzee facilities. It is our hope that the preferences of the chimpanzees, as well as the availability of choices that may encourage species-typical behaviors such as climbing, will assist in promoting the health and behavioral welfare of large groups of chimpanzees in captivity. The changes in space use identified in this study suggest that this large group of chimpanzees adapted positively to a spacious, complex, forested habitat at Chimp Haven in a short period of time.

## 5. Conclusions

The chimpanzees in this study began to use the habitat immediately and continued to increase their use of the space over time, with females being more likely to utilize the arboreal space than the males. In addition, individuals that were utilizing the habitat were more likely to spend time within proximity to each other over time. In particular, they tended to spend more time with unfamiliar group mates towards the end of the study period. These results may be due to cooler temperatures as well as increased familiarity with one another over time. Overall, the chimpanzees spent more time outside in the habitat after the July release. The habitat allowed the chimpanzees to explore and become familiar with an area over time, which may have increased the use of this novel space. 

The authors hope that this study will assist other facilities in their decision-making processes regarding future enclosure design and will encourage them to house chimpanzees in large, natural outdoor spaces which provide a more complex environment. These types of natural spaces provide the chimpanzees with more choice, control, and improved welfare. This study may also provide information to assist in managing larger, complex social groups over time. Ultimately, we believe that improvements in choice and control for chimpanzees lead to improved welfare [61,62]. In this study, the chimpanzees were provided with multiple choices in terms of indoor and outdoor space, increased space, and complex enclosures with natural vegetation, as well as increased choices in social partners. 

## Figures and Tables

**Figure 1 animals-13-00961-f001:**
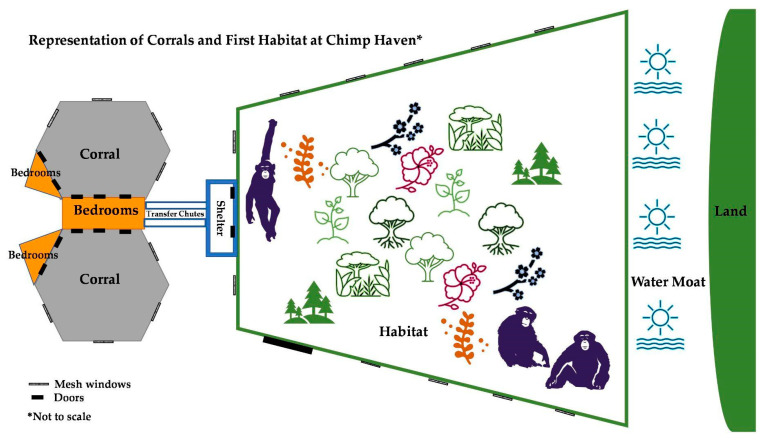
Diagram representing the corrals the chimpanzees had access to throughout the study and the habitat that the chimpanzees had access to beginning in July 2005. Location names are in bold. Diagram is not to scale.

**Figure 2 animals-13-00961-f002:**
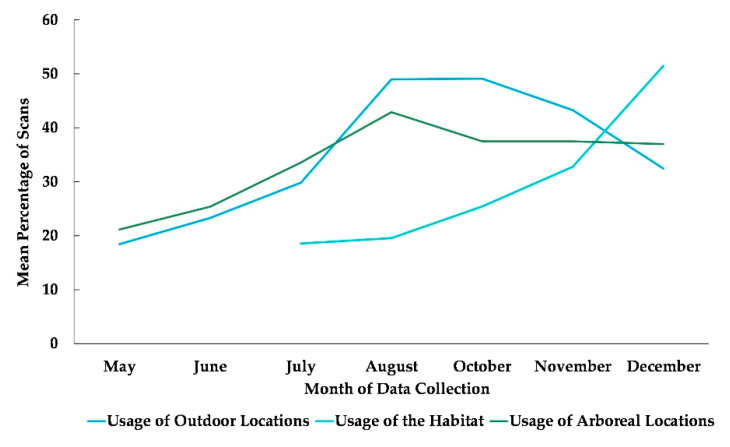
Mean percentage of scans that the chimpanzees used outdoor and arboreal locations from May to December and habitat locations from their July release into the habitat to December. No data collection occurred during the month of September.

**Figure 3 animals-13-00961-f003:**
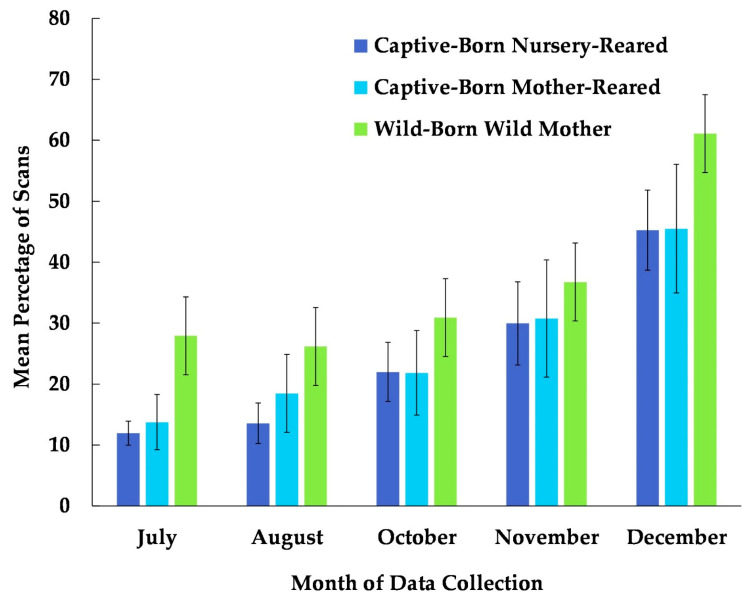
Mean percentage of scans that captive-born nursery-reared, captive-born mother-reared, and wild-born chimpanzees utilized the habitat from July to December. No data collection occurred during the month of September.

**Figure 4 animals-13-00961-f004:**
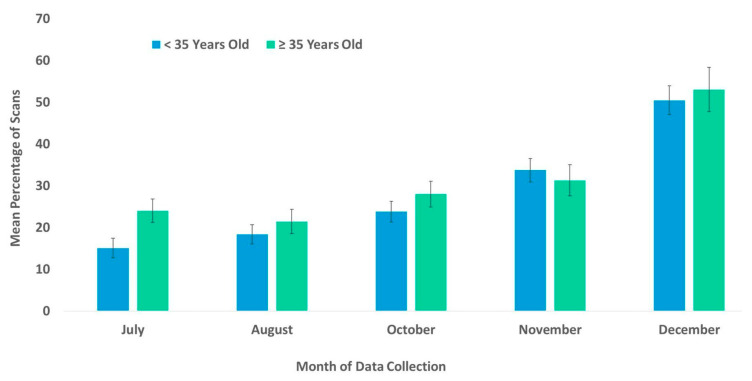
Mean percentage of scans that non-geriatric (<35 years old) and geriatric chimpanzees (≥35 years old) used the habitats from July to December. No data collection occurred during the month of September.

**Figure 5 animals-13-00961-f005:**
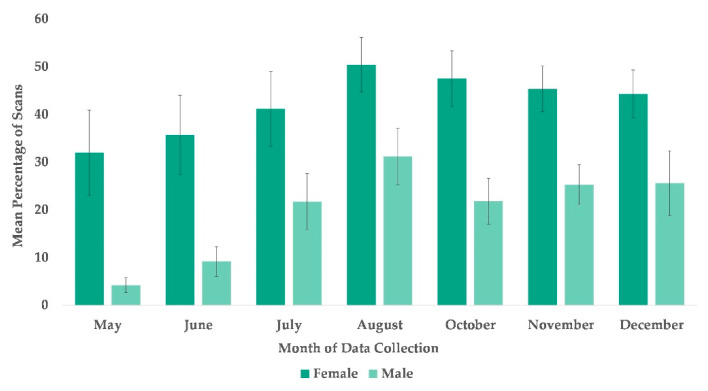
Mean percentage of scans that female and male chimpanzees used arboreal space from May to December. No data collection occurred during the month of September.

**Figure 6 animals-13-00961-f006:**
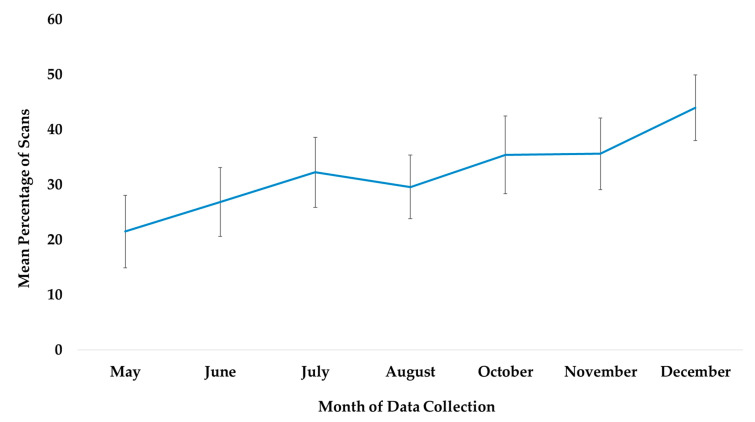
Mean percentage of scans of overall use of indoor arboreal space from May to December. No data collection occurred during the month of September.

**Figure 7 animals-13-00961-f007:**
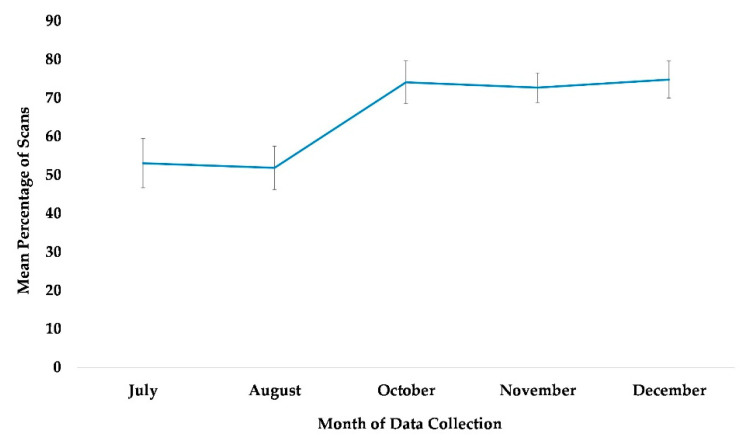
Mean percentage of scans that the chimpanzees used terrestrial space in the habitat from July to December. No data collection occurred in the month of September.

**Figure 8 animals-13-00961-f008:**
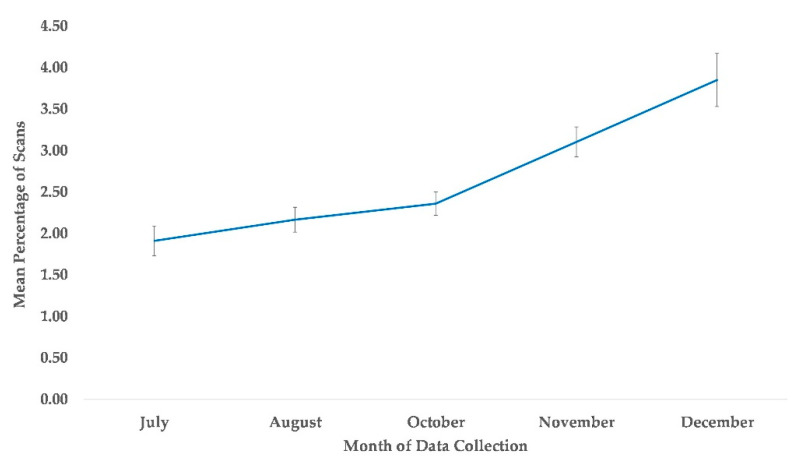
Mean percentage of scans that chimpanzees were in proximity to one another from July to December. No data collection occurred during the month of September.

**Figure 9 animals-13-00961-f009:**
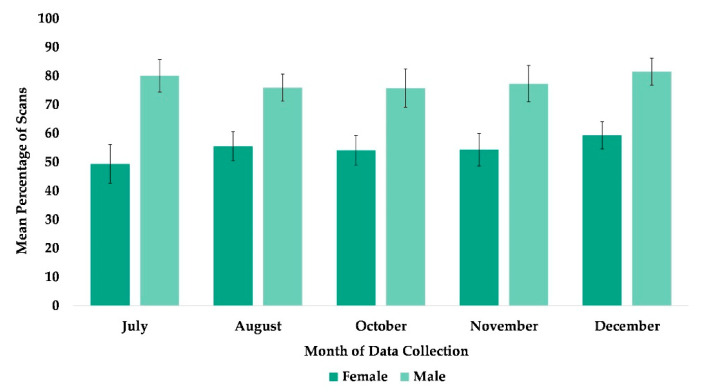
Mean percentage of scans that females and males were in proximity to unfamiliar group members from July to December. No data collection occurred in the month of September.

**Table 1 animals-13-00961-t001:** Chimpanzee, sex, age, and rearing history.

Chimpanzee	Sex	Age	Rearing History
Aug	Female	30	Captive-Born Nursery-Reared
Con	Male	19	Captive-Born Mother-Reared
Dee	Female	19	Wild-Born Wild Mother
Dea	Female	38	Wild-Born Wild Mother
Ell	Female	33	Captive-Born Mother-Reared
Gya	Female	38	Wild-Born Wild Mother
Joh	Male	41	Wild-Born Wild Mother
Jun	Male	37	Captive-Born Nursery-Reared
Lta	Female	46	Wild-Born Wild Mother
Mgn	Male	16	Captive-Born Nursery-Reared
Msn	Male	21	Captive-Born Nursery-Reared
Mrv	Male	29	Captive-Born Nursery-Reared
Pdn	Male	26	Captive-Born Nursery-Reared
Rta	Female	41	Wild-Born Wild Mother
Sra	Female	20	Captive-Born Mother-Reared
Shi	Female	33	Captive-Born Mother-Reared
Sza	Female	29	Captive-Born Nursery-Reared
Tra	Female	41	Wild-Born Wild Mother

**Table 2 animals-13-00961-t002:** Parameter estimates from the GLMMs ^a^.

Prediction	Factor	Factor Level ^b^	Estimate	S.E.	Z	*p* Value	Lower Ci ^c^	Upper Ci ^c^
Use of all outdoor space	Time	May	Reference					
		June *	0.349	0.164	2.13	0.0330	0.028	0.670
		July ***	0.705	0.16	4.4	0.0000	0.391	1.020
		August ***	1.551	0.155	10.03	0.0000	1.248	1.855
		October ***	1.566	0.157	9.99	0.0000	1.259	1.873
		November ***	1.322	0.158	8.36	0.0000	1.012	1.632
		December ***	0.838	0.162	5.17	0.0000	0.521	1.156
Habitat use	Time	July	Reference					
		August	0.0916	0.144	0.64	0.5245	−0.190	0.374
		October **	0.4401	0.14	3.15	0.0016	0.167	0.714
		November ***	0.8047	0.136	5.91	0.0000	0.538	1.072
		December ***	1.623	0.135	12.03	0.0000	1.359	1.888
	Age	(Continuous) *	−0.0396	0.018	−2.27	0.0235	−0.074	−0.005
	Rearing History	Captive-Born Mother-Reared	Reference					
		Captive-Born Nursery-Reared	−0.022	0.332	−0.07	0.9470	−0.672	0.628
		Wild-Born Wild-Mother-Reared **	1.1175	0.388	2.88	0.0040	0.357	1.878
Use of all arboreal space	Time	May	Reference					
		June	0.327	0.19	1.72	0.0853	−0.045	0.699
		July ***	0.8	0.189	4.24	0.0000	0.430	1.170
		August ***	1.282	0.19	6.75	0.0000	0.910	1.654
		October ***	1.02	0.189	5.39	0.0000	0.649	1.391
		November ***	1.032	0.193	5.36	0.0000	0.654	1.409
		December ***	0.968	0.194	4.99	0.0000	0.588	1.348
	Sex	Female	Reference					
		Male **	−1.103	0.359	−3.07	0.0021	−1.807	−0.400
Use of indoor arboreal space	Time	May	Reference					
		June	0.417	0.232	1.79	0.0729	−0.039	0.872
		July **	0.741	0.231	3.2	0.0014	0.288	1.195
		August **	0.625	0.233	2.68	0.0074	0.168	1.082
		October ***	0.936	0.224	4.17	0.0000	0.497	1.376
		November ***	0.957	0.229	4.17	0.0000	0.507	1.406
		December ***	1.354	0.234	5.8	0.0000	0.897	1.812
	Sex	Female	Reference					
		Male *	−1.042	0.49	−2.13	0.0333	−2.002	−0.083
Use of terrestrial space (habitat)	Time	July	Reference					
		August	−0.114	0.255	−0.45	0.6553	−0.613	0.385
		October ***	0.94	0.265	3.55	0.0004	0.421	1.459
		November **	0.884	0.276	3.2	0.0014	0.342	1.425
		December ***	1.048	0.279	3.76	0.0002	0.501	1.594
Proximity to all group members	Time	July	Reference					
		August	0.176	0.114	1.55	0.1220	−0.047	0.399
		October	0.189	0.114	1.66	0.0960	−0.034	0.411
		November ***	0.493	0.107	4.6	0.0000	0.283	0.703
		December ***	0.717	0.104	6.93	0.0000	0.514	0.920
	Sex	Female	Reference					
		Male *	−0.247	0.101	−2.46	0.0140	−0.444	−0.050
Proximity to unfamiliar group members	Time	July	Reference					
		August	0.086	0.132	0.65	0.5130	−0.172	0.344
		October	0.039	0.131	0.29	0.7680	−0.217	0.294
		November	0.077	0.13	0.59	0.5519	−0.178	0.332
		December *	0.31	0.133	2.33	0.0198	0.049	0.570
	Sex	Female	Reference					
		Male ***	1.186	0.351	3.38	0.0007	0.499	1.874

^a^. Factors in the best fit model for each prediction based on AIC and Delta AIC values are listed in this table. ^b^. * *p* ≤ 0.05; ** *p* ≤ 0.01; *** *p* ≤ 0.001. ^c^. Confidence intervals at 95%.

## Data Availability

The data presented in this study are available on request from the corresponding author. The data are not publicly available due to restrictions regarding data sharing.

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
