# Peer review of "How Sanctuary Chimpanzees (Pan troglodytes) Use Space after Being Introduced to a Large Outdoor Habitat"

_animals, 2023, doi:10.3390/ani13060961_

Round 1

Reviewer 1 Report

General comments: The manuscript entitled “How sanctuary chimpanzees (Pan troglodytes) use space after being introduced to a large outdoor habitat” by Amy Fultz et al. addresses an interesting topic: how the integration of a small group of chimpanzees into a larger social group and their introduction in a larger outdoor enclosure can affect their use of space. The topic is very interesting, especially from an applicative point of view. The impact that the fusion of two group of chimpanzees and how they use the outdoor enclosure can have an implication in the captive management of this species. However, I have some major concerns about methods and data analysis. Due to the possible impact that this work can have in academia, I encourage the authors to make a fine revision in order to solve major concerns.

Major concerns:

Data collection was conducted via instantaneous scan sampling. Even if the study was conducted over a total of seven months, during both daytime and nighttime (20 hours per day), the authors have collected only one scan per hour (16 scan per day). Even if I agree with the authors that instantaneous scan sampling can be useful to collect data on the use of space, I think that one scan per hour can be not enough to conduct a fine investigation of the use of space, especially with a species that in the wild can cover territories of up to 300 km2. Moreover, it is possible that not all subjects were visible in all the scans. Thus, I would like to know the total number of scans collected (±SD) during the entire data collection and for each month, as well as the average number of scans per individual (±SD), in order to evaluate if the amount of data collected could be considered enough for the purpose of the study.

Since data were collected by “nine trained individuals”, I would like to know if the authors have measured an inter-observer reliability, in order to check whether all observers collected and entered data in the same manner.

I agree with the method used to calculate the percentages regarding chimpanzees' use of space. I am a little confused on how the percentages about the proximity between individuals were calculated (lines 265-271). I suggest to calculate the percentage of scans in which one individual (individual A) is in proximity to another (individual B) by dividing the number of scans in which one subject (individual A) is in proximity to at least another one by the total number of scans of that subject (individual A): (number of scan where the individual A is proximity with at least another one)/(total number of scans of the individual A). As regards the percentage of scan where one individual is in proximity with an unfamiliar/familiar subject, I suggest to divide the number of scan where the subject (individual A) is proximity with an unfamiliar/familiar subject (individual B) by the total number of scan where the subject is in proximity with another subject (unfamiliar and/or unfamiliar): (number of scan where the individual A is proximity with an unfamiliar/familiar subject)/(total number of scans where the individual A is proximity with unfamiliar and/or familiar subject). Moreover, since via scan sampling is not possible to report the direction of proximity (if it is individual A that gets into proximity with individual B or vice versa), I would like to know if the authors considered the dyads subjectA/subjectB and subjectB/subjectA as two different dyads or as the same. I request this information because, due the fact that proximity has no direction, if the authors have considered the dyads subjectA/subjectB and subjectB/subjectA as two different dyads, data are duplicated.

I agree with the authors that the use of outdoor/indoor enclosure could be affected by the temperature. Thus, I was wondering if it would be possible to include within the models the average monthly temperature as fixed factor.

As regards the results section, I noticed that the p values reported in the table do not match those reported in various sections of the text. Is this a typo? Moreover, the authors correctly applied the Bonferroni correction, so the critical value is 0.007. Nevertheless, in some cases they reported that some variables had a significant effect even if their probability was 0.038 (for example the probability of the variable subjects’ age in the section 3.2 Habitat Use). Is this because the value reported is already adjusted for the Bonferroni correction?

Finally, I think that the discussion section can be improved providing more comprehensive explanations of the study results. For example, at lines 452-456 the text says: “Proximity to group mates may change due to available space, housing conditions or familiarity with group members. We could not determine whether or not this tendency was due to the new group becoming more stable over time or the increase in space as both occurred concurrently”. I encourage the authors to discuss the results with more than one non-exclusive possible explanations.

Minor concerns (by line number):

Line 27: I think there was a conversion error.

Line 47: [1-5] instead of [1,2,3,4,5].

Line 54: same as line 47.

Line 61: replace "m2" with "m2".

Line 63: same as line 61.

Line 77: same as line 47.

Line 104: same as line 47.

Line 118: use [ ] instead of { }.

Line 129: same as 61.

Line 154: I believe that dividing and numbering the predictions makes them clearer and more intuitive.

Line 162: please add some references.

Line 171: please add some references.

Line 186: I think there was a conversion error.

Line 191: figure 2?

Line 218: replace “2.3” instead of “1.3”.

Line 226: please add some references (for example Altmann, J. Observational study of behavior: Sampling methods. Behaviour 1974, 49, 227–266. https://doi.org/10.1163/156853974X00534).

Line 242: I agree that proximity and contact sitting could be included in the same behavioral category, but I would prefer that the authors keep the definitions separate. Thus, proximity is defined as being within arm’s reach and contact sitting is defined as being in contact with another subject.

Line 244-246: can the authors add a reference to justify the 6-week time window to divide between familiar and unfamiliar?

Line 247: replace “2.4” instead of “1.3”.

Line 250-251: I think that this is a repetition.

Line 262-264: please, can you rephrase this sentence?

Line 292: please use the same font as the rest of the manuscript.

Table 1: where is the variable sex in the section about “Use of terrestrial space (habitat)”?

Line 301: replace "df1" with "df1" (the same for all the other cases).

Line 305: you mentioned October, November and December, but August?

Fig 1: can you specify if the usage of arboreal locations is indoor or outdoor?

Fig 2: replace "December.." with "December.".

Fig 3: can you add the error bars? Moreover, can the authors add a reference to justify the division between geriatric and non-geriatric?

Line 328: replace “3.3” instead of “1.3”.

Line 335: the authors cite the Figure 1, but in Figure 1 there is nothing that informs readers about subjects’ sex.

Line 339: can you add a representative figure?

Line 340: replace “3.4” instead of “1.3”.

Line 346: the same as for line 339.

Line 347: replace “3.5” instead of “1.3”.

Line 357-363: can you add a representative figure?

Line 368: please add the conversion into °C.

Line 412: replace “]” instead of “}”.

Line 346-347: I think that this sentence could be more useful at the beginning of the section.

Line 443-446: why?

Line 487: I think this part, as it is written now, is more a summary of the results than a conclusion. The conclusion section should provide a "take home" message and outline the importance of your study.

Line 524: the Reference section must be improved. There is a lot of citations without DOI, some abbreviations are incorrect, and there are other formatting inaccuracies.

Reviewer 2 Report

The article “How Sanctuary Chimpanzees (Pan troglodytes) Use Space After Being Introduced to A Large Outdoor Habitat” presented by the authors is definitely a very interesting study worth publishing in this journal. I feel that this type of studies is very much needed and want to encourage the authors to provide more studies related to this topic in the future. I also want to congratulate them on conducting long-term observations that are difficult to achieve due to the resources and coordination needed. I tried my best to provide helpful suggestions to further improve this manuscript. I don´t think that my suggestions will be difficult or problematic for the authors to take into consideration when modifying the current version. However, it might take them some time. In terms of quality I would suggest to rate the required modifications as “minor revision”, yet I believe they might need more than 5 days typically granted for a minor revision. Hence, I suggest to provide them with more time although I don´t criticize to the extent of calling it a “major revision”.

As a general observation I believe the author may consider being a bit more daring, referring to some of the observed increases in space use and social proximity as indicators of improved general wellbeing of the chimpanzees. Furthermore I would welcome them to state some clear statements regarding their opinion of said wellbeing improvement and how they suggest other housing organizations could learn and take advantage from their findings.

Following, please find my suggestions:

At some point a formatting issue seems to have appeared as the chapter and section numbering in the file are not correct. There are many section marked as “1.3” and the order does not make sense. But this should be fixed very easily.

Throughout the manuscript (but mostly in the simple summary and abstract) both the enclosures as well as group sizes are repeatedly referred to as “large” y suggest changing some “large” for other words such as “extensive” or “spacious” to avoid word repetition.

Simple summary: I suggest to add one last phrase to highlight the importance of looking into “arboreal and terrestrial use” as well as why measuring social proximity is of interest

Line27: 20.16m2 only refers to the tree/forest area, but how big is the whole of the new outside habitat. Also, later on I see that when you refer to “arboreal” that you not only refer to “actual ON TREES” but any elevated positioning (including indoor platforms etc.) I find it somehow confusing. It might be a good idea not to use the word arboreal but use a more generic term like “off-ground” or “in elevation”. (but this might be me not being an English native speaker)

Line 37: when you say “outside space” is that the same as “terrestrial space”, i.e. “not arboreal space”?

Line 41: if the abstract word count permits it, it might be nice to highlight the utility for management purposes and importance for research to conduct such longitudinal observations and evaluations.

Intro:

Line 45: I suggest to change “fluid” for “flexible”

Line 56: As you explain the climate conditions of their natural habitat, I suggest adding a phrase summarizing climate variation affecting the sanctuary site to allow the reader to better understand your climatic conditions. (you eventually do that in the discussion, but I think it would improve the readability to also mention this here)

Line 60: The definition of “adult chimpanzees over 10kg” struck me as weird, as I have never seen an adult chimpanzee weighing less than 30kg in captivity, and that would already be considered extremely small. If not strictly necessary I suggest to eliminate the “over 10kg”.

Line 162: Regarding your hypothesis regarding the arboreal use based on the weight differences between sexes: I suggest you add average weights + Standard deviation for both sexes here to make your reasoning more obvious. (also in future studies you might want to add the chimpanzees weight as a predictor as your reasoning sounds very plausible)

Line 168: there is an extra “space” between “predicted” and “that”

Line 168: when referring to “oldest” I suggest you provide a brief age definition and/or average age for the “eldest” (later I see you used the age as a continuous variable but in the graphics used a definition of two categories “under 35 vs 35+”)

Methods:

Facilities: This might be due to my lacking English level, but I was surprised by the use of the word “corral” and I fail to understand the difference between “corral” and “outdoor habitat” that was added in Month3. I am not refereeing to the explained differences in material, size and arboreal possibilities. Just wondering why, one is called a corral and the other is called outdoor habitat. Is this simple based on the material used for limiting the area (concrete wall vs. fencing)?

Line 179-198: Although I welcome the detailed explanation of measurements (should definitely remain) it also is quite difficult to follow. I would suggest to add a drawn map to make it easier for the reader to understand the distribution and measurements.

Line 186: I think 217,800f2 should actually be 217.8f2 if I am not wrong as that would equal the 20.16m2

Line 191: You mention a Figure 2 here. This seem to be an error I assume, as (a) there was no previous mention of a “Figure 1” and (b) further down Figure 2 in the manuscript is about “Mean percentage of scans that captive born nursery reared, captive born mother reared and wild born chimpanzees utilized the habitat from July to December..”.

Line 214-217: I would welcome a Table here with columns regarding the age, sex, rearing condition and even “original Small group”/familiarity.

Line 220: eliminate extra space between “collection” and “was”

Line 223: Did the nine trained observers undergo an inter-observer reliability test: This would be ideal and if so should be stated. Although I understand that the data collection was a quite simple and straight forward one.

Line 225-228: Regarding the schedule am I correct to assume that this data was not recorded directly but taken from video or picture footage? If so than please specify.

Line 241: when you define proximity, could you specify that “being within arm’s reach of or touching another group member” does refer to both active social interactions as well as the passive state of simply being within arm’s reach of or touching another group member.

Line 272 Statistical Analysis: I tend to use and prefer different ways of GLMM model selection, analysis and interpretation, but this type of analysis can be applied in many different ways and from what I can see the one used here seems to be correct. However, if I understood correctly the authors tested their dependent variables for normality and then transformed non-normal distributed variables. In continuation they conducted LMMs with a normal distribution. All this is fine but in that case the naming is not correct. It should be called LMM and not GLMM as far as I understand.

I would suggest to add a Table in the supplementary material providing the AIC, Delta AIC, and used predictors/fixed factors of all selected best models, so readers can consult these values.

Line 292: there is a font change that needs to be fixed

Results and last bit of Statistical Analysis:

Line 290: You say you present a total of 8 models (each with a different dependent variable) but in the results Table 1 there are only 7 models, which means one did not provide any significant results. Please still state that model, as no result is also information that can be useful and should be presented in the results and discussed later on.

Table1: I was wondering why you used the age variable in a continuous form if later in the Figure you split the animals in two categories (above or below 35). Not saying to change it now, but for the future you might try simply creating age categories (for example, adult vs senior) and use this binary variable in the LMM.

Line 289-293: I am not sure what is being explained here. I would assume that this refers to post hoc testing of the variables that were significant in the LMM models. However, it doesn´t state what type of test was used, but only what type of correction (Bonferroni) was applied.

Regarding the follow up analysis and the results that are being presented within the text; I do miss a bit of information to fully understand those. Could you add a table with all these results in the supplementary material? For example: “Only the variable time was significantly associated with the subjects’ use of outdoor locations (F = 39.323, df1 = 6, df2 = 112, 301 P < 0.0005).” I ausme that might have been an anova test (typeIII) but I lack that information due to missing explanations.

Also, Table 1 indicated that 7 of the 8 models always had at least 1 predictor giving a significant result however in the written sub sections following you missed to provide a section regarding “Use of indoor arboreal space”.

Although I like the graphics provided, I would suggest to add confidence interval plots for all significant variables as to show the tendencies and differences between the Factor levels of each significant factor.

If that would be done than I would also suggest to move Table 1 to the supplementary material as it breaks the reading flow, being so extensive.

Discussion:

Line 365 I would welcome a brief summary of the findings before starting to discuss each section separately.

Line 365+ Use of all outdoor space:

You mainly argue here that the increase over time is likely due to season/temperature. I do agree with this statement as I also believe there might be a big influence due to temperature. However, your original statement/idea was that they would use the outdoor space more over time as the got more familiar with the terrain and new group members etc., yet you barely mention this here anymore.

Line 410: eliminate extra space after “groups”

Line 401+ Habitat use and Use of Abboreal space: Although I agree with all statements here I feel there might be to little emphasis of the success regarding the chimpanzees effectively taking advantage of the extra space and new additional arboreal/3D climbing possibilities the sanctuary provided.

Line 447 Proximity to Group Members:

The increase over time here could also be partly due to the decreasing temperature (you could briefly mention that). On the contrary if time spent in proximity increases over time regardless of the familiarity of the close by chimpanzee, this still seems a clear sign of improved association. Especially if you compare with the first observations where the available space was smaller, i.e. in terms of probability before having access to the spacious outdoor installations it should statistically be more likely to have seen them in proximity.

Line 484-486: Perhaps you could push even a bit more and suggest that the chimpanzee’s welfare as been improved by (a) a bigger group and (b) a more spacious and naturalistic enclosure?

Although I made quite a few suggestions I want to make sure the Editor and author understand that I am very impressed by this study and very much look forward to see this published in this Journal. Congratulations once again.

Round 2

Reviewer 1 Report

Dear authors,

Thank you for fully answering my questions. 

I think now the manuscript just requires a final read to check for typos and formatting errors.

Line 331-334: please remove bold

Line 341: please remove italics 

Line 342: there is a double )). Remove one

Line 385: please be consistency with the style of the different paragrphs

Line 398: replace "Figure 7" instead of "Figure 7." 

Line 414: same as line 385

Line 710-711: remove these lines

All references: please try to be consistency with the DOI. You can use DOI:number or http://..... not a mix of the two.
